# Screening of Ionic Liquids against Bamboo Mildew and Its Inhibition Mechanism

**DOI:** 10.3390/molecules28083432

**Published:** 2023-04-13

**Authors:** Chunlin Liu, Shiqin Chen, Yingying Shan, Chungui Du, Jiawei Zhu, Qichao Bao, Yuran Shao, Wenxiu Yin, Fei Yang, Ying Ran, Yuting Wang

**Affiliations:** College of Chemistry and Materials Engineering, Zhejiang A & F University, Hangzhou 311300, China; eustaceweaver7187@gmail.com (C.L.);

**Keywords:** ionic liquid, antifungal properties, antifungal mechanism, mildew

## Abstract

Ionic liquids are a class of organic molten salts that consist entirely of cations and anions. They are characterized by their low vapor pressure, low viscosity, low toxicity, high thermal stability, and strong antifungal potential. In this study, the inhibitory performance of ionic liquid cations against Penicillium citrinum, Trichoderma viride, and Aspergillus niger was investigated, along with the mechanism of cell membrane disruption. The Oxford cup method, SEM, and TEM were employed to examine the extent of damage and the specific site of action of ionic liquids on the mycelium and cell structure of these fungi. The results showed that 1-decyl-3-methylimidazole had a strong inhibitory effect on *TV*; benzyldimethyldodecylammonium chloride had a weak inhibitory effect on *PC*, *TV*, *AN*, and a mixed culture; while dodecylpyridinium chloride exhibited significant inhibitory effects on *PC*, *TV*, *AN*, and Mix, with more prominent effects observed on *AN* and Mix, exhibiting MIC values of 5.37 mg/mL, 5.05 mg/mL, 5.10 mg/mL, and 5.23 mg/mL, respectively. The mycelium of the mildews showed drying, partial loss, distortion, and uneven thickness. The cell structure showed separation of the plasma wall. The absorbance of the extracellular fluid of *PC* and *TV* reached the maximum after 30 min, while that of *AN* reached the maximum after 60 min. The pH of the extracellular fluid decreased initially and then increased within 60 min, followed by a continuous decrease. These findings provide important insights for the application of ionic liquid antifungal agents in bamboo, medicine, and food.

## 1. Introduction

In recent years, ionic liquids have garnered significant attention as a “green solvent” due to their unique physicochemical properties, such as low vapor pressure [1], high thermal stability [2], high solubility [3], high viscosity [4], and tunable structures. These organic molten salts are composed of cations and anions and have a melting point below 100 °C [5,6]. Owing to their versatility, ionic liquids are widely used as catalysts in chemical synthesis [7], electrolytes in supercapacitors [8], and antimicrobial agents in pharmaceutical and food industries [9,10]. Since the onset of the COVID-19 pandemic, there has been an increased interest in researching antimicrobial agents [11,12,13], leading to the investigation of ionic liquids as effective agents for the biomedical industry. The majority of studies exploring the antimicrobial potential of ionic liquids have focused on imidazoles, pyridines, and quaternary ammonium salts, all of which possess strong bactericidal properties [14]. These compounds have attracted significant attention from researchers, with a considerable number being commercially available, low in toxicity, and thermally stable, indicating their potential for broad applications in antifungal research.

Currently, research on the antifungal potential and inhibition mechanisms of imidazole, pyridine, and quaternary ammonium salts of ionic liquids is predominantly focused on bacteria such as Staphylococcus aureus, Escherichia coli, Bacillus subtilis [15,16], and fungi such as Candida albicans, Aspergillus flavus, and Fusarium acnes [17,18,19]. However, reports on the antifungal properties and mechanisms of these ionic liquids in other fields are scarce, necessitating further exploration and expansion of their research and application fields [20].

Bamboo, a rapidly growing plant, possesses a variety of desirable properties such as degradability, renewability, high tensile strength, toughness, and low cost [21,22]. As a natural material, bamboo and its products are widely used in diverse applications, including construction, furniture, decoration, food, packaging, and gardening [23,24,25]. However, bamboo and its products are highly susceptible to mildew growth caused by fungi such as Penicillium citrinum, Trichoderma viride, and Aspergillus niger in warm and humid environments. The factors contributing to bamboo mildew growth can be attributed to the high content of nutrients, such as starch, sugar, and protein, present in bamboo, which provide a suitable substrate for mildew growth [26,27], as well as the external environmental conditions of high humidity (≥93%) and temperature (20–30 °C), which favor mildew growth on bamboo surfaces [28,29]. Notably, the black spots resulting from mildew contamination are difficult to remove by conventional methods, resulting in significant economic losses and resource wastage [28,30]. Therefore, it is essential to explore novel, safe, and long-lasting mildew inhibitors for the prevention and control of bamboo and its products, which is crucial for reducing the economic impact of bamboo contamination with mildew.

In recent years, there has been significant progress in the development of bamboo mildew inhibitors. Research has focused on the mechanisms underlying bamboo mildew prevention, the efficacy of mildew prevention, and the improvement of mildew prevention processes. The number of reports on this topic has increased [31,32]. Currently, chemical mildew inhibitors such as tebuconazole, propiconazole, and organic iodide are widely used, but they often provide only short-term protection and can be harmful to the environment and human health due to their high toxicity [33]. Recent studies have explored the use of ionic liquids for bamboo mildew prevention. For example, Benedetto et al. [34,35] found that the cationic structure of choline-based and imidazole-based ionic liquids can penetrate irreversibly into the lipid layer of the biofilm, accounting for about 2–10% of the bilayer volume. Dani et al. [36] verified the interfacial behavior of surface-active ionic liquids (SAILs) in combination with biocidal and cytotoxic assessments. The results revealed that SAILs with the alkyl chain-length greater than C-8- act as a fair antimicrobial agent against the selected microbial strain that is attributed to the enhanced degree of SAILs’ hydrophobicity. In Zhang et al. [19], the antibacterial, antifungal, and anticancer properties of β-pinene quaternary ammonium salts were designed, synthesized, and evaluated. The results showed that bis-hydronopyl dimethyl ammonium chloride and bis-hydronopyl dimethyl ammonium iodide presented remarkable antimicrobial activity against the tested fungi and bacteria. Rezki et al. [37] investigated a number of pyridine-based ionic liquids—novel inhibitors of fungal ergosterol biosynthesis—and the results showed that some of these pyridine-based ionic liquids have significant anti-Candida activity, possibly by interacting with ergosterol in the fungal cell membrane to reduce its content and ultimately lead to membrane damage. Trush et al. [38] studied the ecotoxicity and degradability of cationic fungicides with ester-functionalized pyridinium ionic liquids. The results showed that both ester-functionalized ILs and cetylpyridinium chloride (CPC) showed significantly reduced antibacterial activity compared to 1-dodecylpyridinium chloride (PyrC(12)-Cl), but the ester-functionalized ILs exhibited excellent antifungal activity against Candida albicans. However, there are few studies on the inhibition performance and mechanism of inhibition of *PC*, *TV*, *AN*, and their mixed mildews using imidazole, quaternary ammonium salts, and pyridine ionic liquids. Therefore, this work employs these three types of ionic liquids to investigate mildew prevention in bamboo and provides a theoretical reference for the research and application of bamboo anti-mildew agents.

## 2. Results and Analysis

### 2.1. Oxford Cup Method to Investigate the Inhibition Performance of Ionic Liquids against Bamboo Mildew

Based on previous research, bamboo mildew is mainly caused by three fungi, *PC*, *TV*, and *AN* [39,40,41]. The Oxford cup method was employed to investigate the inhibitory activity of various ionic liquids at different concentrations against the growth of bamboo mildew caused by *PC*, *TV*, *AN*, and a mixture of these fungi. The strength of inhibition was characterized by measuring the diameter of the inhibition zone, and three ionic liquids, namely 1-decyl-3-methylimidazole chloride, benzyldimethyldodecylammonium chloride, and dodecylpyridinium chloride, were evaluated. The goal was to identify the ionic liquid with the strongest inhibitory effect on bamboo mildew. The results of the inhibitory activity of the four aforementioned bamboo mildew fungi, after being incubated in an artificial climate chamber for 2–3 days, are presented in Figure 1, Figure 2, Figure 3 and Figure 4.

Figure 1a demonstrates that with the increase in concentration of 1-decyl-3-methylimidazole chloride solution from 2.5 mg/mL to 40 mg/mL, the inhibition zone of each mildew exhibited a corresponding increase in diameter, ranging from a minimum of 8 mm to a maximum of 46 mm. Notably, at the same concentration of 1-decyl-3-methylimidazole chloride, the diameter of the inhibition zone of Mix was significantly smaller when compared to those of *TV* and *PC*. The inhibition rate table in Figure 1b shows that 100% inhibition of *PC*, *TV*, *AN,* and Mix was achieved when the concentration of 1-decyl-3-methylimidazole chloride was at 20 mg/mL.

Figure 2a depicts that the diameter of the inhibition zones of *TV* and Mix treated with benzyldimethyldodecylammonium chloride surpassed that of *PC* and *AN*. Additionally, the diameter of the inhibition zones of *PC*, *AN*, and Mix showed no significant change upon increasing the concentration of the solution to 10 mg/mL. However, the diameter of the inhibition zones of all mildews increased significantly when the concentration was further increased to 20 mg/mL. The inhibition rate table in Figure 2b shows that 100% inhibition of *PC*, *TV*, *AN,* and Mix was achieved when the concentration of benzyldimethyldodecylammonium chloride was at 20 mg/mL.

In Figure 3a, *PC*, *TV*, *AN*, and Mix treated with dodecylpyridinium chloride exhibited noticeable and uniformly sized inhibition zones. The inhibition rate table in Figure 3b shows that 100% inhibition of *PC*, *TV*, *AN*, and Mix was achieved when the concentration of dodecylpyridinium chloride was at 10 mg/mL.

It can be seen from Figure 4 that at the same concentration, 1-decyl-3-methylimidazolium chloride in Figure 4a showed a larger circle of inhibition against *TV* at the same concentration; benzyl dimethyl ammonium chloride in Figure 4b showed a smaller circle of inhibition against all strains tested; dodecylpyridinium chloride in Figure 4c showed a larger circle of inhibition against all strains tested, with an inhibition diameter of 17 mm against *AN* at 10 mg/mL.

Based on the above inhibition activity test, it can be inferred that 1-decyl-3-methylimidazole chloride, benzyldimethyldodecylammonium chloride, and dodecylpyridinium chloride exhibited varying degrees of inhibition effects on *PC*, *TV*, *AN*, and Mix under identical concentration conditions. Finally, dodecylpyridinium chloride demonstrated a better inhibition effect on *PC* and *TV*, while its inhibition effect on *AN* and Mix was more significant than the first two ionic liquids. Overall, dodecylpyridinium chloride is a more suitable antimicrobial agent, exhibiting a better inhibitory effect on bamboo mildew.

### 2.2. Analysis of MIC and MFC Values of Bamboo Mildew

Using the multiplicative dilution method, we determined the MIC and MFC of various types of ionic liquids at different concentrations against *PC*, *TV*, *AN*, and Mix mildew. The results of these tests are presented in Table 1.

Ionic liquids possess high viscosity [4] properties that enable them to adhere to the surface of cell membranes, facilitating the insertion of hydrophobic alkyl chains into the cell membrane. The length of the alkyl chain determines the extent of inhibition, with longer chains exhibiting greater inhibition and requiring lower concentrations of the ionic liquid for the same effect [42]. Table 1 demonstrates that dodecylpyridinium chloride has a smaller MIC value for bamboo mildew than 1-decyl-3-methylimidazole chloride and benzyldimethyldodecylammonium chloride. Furthermore, the MIC results indicate that dodecylpyridinium chloride exhibits a better inhibition of *TV* and *AN* growth by having a smaller MIC value for these fungi. Although dodecylpyridinium chloride has a superior inhibitory effect on bamboo mildew, its MIC value for Mix is not greater than the MIC value for *PC* inhibition. This can be attributed to the less variable environment with a competitive relationship among *PC*, *TV*, and *AN*. *TV* and *AN* exhibit stronger growth in the early stages and consume a portion of the dodecylpyridinium chloride drug and nutrients, resulting in a subsequent reduction in the amount of dodecylpyridinium chloride agent exposed to *PC* growth to a certain degree. This is consistent with the findings of Misra et al. [43] regarding inhibition studies.

Table 1 shows that 1-decyl-3-methylimidazolium chloride exhibited a higher MFC value in the mixed microfungal environment of Mix, which indicates the need for a higher concentration to effectively eradicate the mildews in the complex environment. Benzyldimethyldodecylammonium chloride, on the other hand, exhibited a lower MFC value than *AN* in the Mix environment, likely due to *AN* not gaining a growth advantage in competition. Dodecylpyridinium chloride, while demonstrating a slightly higher MFC value than the other three microfungi, displayed better inhibition against *PC*, *TV*, and *AN* in the same environment. As a result, dodecylpyridinium chloride may serve as a viable reagent for future studies.

### 2.3. Effect of Ionic Liquid on Mycelial Morphology of Bamboo Mycorrhizal Fungi

Based on the MIC and MFC test results of various types and concentrations of antimicrobial agents against bamboo mildew, dodecylpyridinium chloride was selected as the preferred ionic liquid for subsequent studies. The MIC and MFC concentrations of dodecylpyridinium chloride were then applied to treat *PC*, *TV*, and *AN*.

The mycelial structure is a crucial component of microfungal cells, serving as the “tentacle” of their cell structure. When agents or external factors damage the mycelium, the cell wall and intracellular enzymes, physicochemical structures, membrane permeability, and nucleic acids of the microfungal cell are all impacted, ultimately leading to cell inactivity [44,45,46]. According to the observations depicted in Figure 5, the mycelium of *PC* displays a well-developed and substantial structure, characterized by a scaly sheath-like surface in Figure 5(a1). In contrast, the mycelium of *TV* exhibits a straight and irregularly patterned surface in Figure 5(b1). Moreover, the mycelium of *AN* displays a rounded and thick surface with a smooth appearance in Figure 5(c1). Due to variations in the content of polysaccharides, proteins, and other polymers comprising the cell wall structure, *PC* and *TV* displayed a striated structure on the surface of their mycelia, while *AN* exhibited a smooth surface structure [47,48]. In the MIC treatment group, *PC* mycelia showed depressions, partial absences, thickening in the middle and thinning at both ends, and distorted shapes in Figure 5(a2); *TV* mycelia exhibited dryness, partial absence, and a distorted shape in Figure 5(b2); *AN* mycelia appeared dry, depressed, and distorted in Figure 5(c2). In the MFC treatment group, *PC* mycelia exhibited surface depressions, extensive dryness, and disorganized growth in Figure 5(a3); *TV* mycelia displayed surface distortion, depression, increased mycelial absence, severe dryness, and disorganized growth in Figure 5(b3); *AN* mycelia appeared severely dry with a slightly distorted shape in Figure 5(c3). The MIC and MFC concentrations of dodecylpyridinium chloride caused basic deformation of the mycelial structures of *PC* and *TV* and disrupted their structural integrity, rendering them biologically inactive. The degree of destruction increased in proportion to the concentration of dodecylpyridinium chloride. This structural damage to the mycelium was attributed to the hydrophobic structure of the long alkyl chain on the dodecylpyridinium chloride cation, which interacted with the titin in the mycelial cell wall through the alkyl chain tail.

The positively charged cation tail was then inserted into the negatively charged lipid bilayer of the mycelium, causing localized fracture damage to the mycelial membrane layer and leading to the leakage of materials from the mycelium to the extracellular environment, ultimately resulting in inactivation [49,50]. Importantly, in the MIC and MFC treatment groups, *AN* mycelia only appeared dry, which may be due to the complex cubic dense structure of the mycelium composed of multiple layers of extracellular matrix, preventing the long alkyl chains on the dodecylpyridinium chloride cation from easily penetrating the membrane of the mycelium [48,51]. Thus, while the surface layer of *AN* mycelia remained smooth and strong, the permeability of the membrane was still disrupted, resulting in the mycelia only appearing to dry out under higher concentrations of dodecylpyridinium chloride treatment. In conclusion, a certain concentration of dodecylpyridinium chloride can lead to the depression, uneven thickness, twisting, disorganization, drying out, and cracking of bamboo mildew mycelia, thereby achieving inhibitory or bactericidal effects.

### 2.4. Effect of Ionic Liquids on the Cytoarchitecture of Bamboo Mycorrhizal Fungi

The MIC and MFC of dodecylpyridinium chloride were utilized to treat *PC*, *TV*, and *AN*. TEM was employed to investigate potential alterations in the cell structure of these three microfungi following treatment with the aforementioned concentrations, in contrast to the control group. Additionally, differences in the extent of cell structure destruction between the MIC and MFC concentrations were examined. The results are presented in Figure 6.

The cell walls and membranes of fungi play vital roles in maintaining the structural integrity, antimicrobial activity, and immune functions of cells [52]. The layered structure of cell walls is mainly composed of β-(1,3) glucan, proteins, and chitin. In addition, aerial hyphae and microfungal conidia are covered by hydrophobic proteins known as hydrophobins, which form small strips that safeguard the spores from enzymes, oxidants, and phagocytes seeking nourishment [53]. The cell membrane, consisting primarily of phospholipids, proteins, and polysaccharides, is an essential barrier that helps maintain the intracellular environment. The integrity of the cell membrane is crucial, as any disruption can be detrimental to the cell’s normal function.

In the control group, Figure 6(a1,b1,c1) show an intact cytoarchitecture, smooth membrane surface, regular shape, and uniform distribution of organelles in *PC*, *TV*, and *AN* without significant mass wall separation. However, in the MIC treatment group, Figure 6(a2,b2,c2) show localized plasma wall separation, uneven membrane surface, and irregular cytoplasmic distribution in *PC*, *TV*, and *AN*. Moreover, in the MFC-treated group, Figure 6(a3,b3,c3) show complete plasma wall separation, cytoplasmic extravasation, and diminished color in the membrane of *AN*, while the membrane disappeared in *TV* and the membrane surface folded in *PC*. The concentration of dodecylpyridinium chloride was directly proportional to its effect on the destruction of the mildew cell structure. As a fungi, bamboo mildew has an outer cell wall, which impedes external substances, including ionic liquids, from entering the cell. Nevertheless, the precise mechanism of action for the interaction of ionic liquids with the proteins on the cell wall is not yet clear [52]. Dodecylpyridinium chloride, which enters the cell membrane of bamboo mycorrhizal fungi through its alkyl chain tail, disturbs the arrangement of phospholipid bilayers, leading to a gradual loss of intracellular substances and cell inactivity. Thus, the mechanism of inhibition of dodecylpyridinium chloride is primarily due to its disruption of the physiological function of the cell membrane, impeding the synthesis of intracellular ergosterol, and ultimately leading to the loss of cytoplasm in the mildew.

### 2.5. Effect of Ionic Liquids on the Release of Cellular Components from Bamboo Mildew

The extracellular solutions of *PC*, *TV*, and *AN*, treated with varying concentrations of dodecylpyridinium chloride for 0, 30, 60, 90, and 120 min, were subjected to UV spectrophotometry for detection of their UV absorbance values at 260 nm. The results of these measurements are depicted in Figure 7.

The study demonstrated that the absorption of UV light by bamboo mildews was notably intensified at a wavelength of 260 nm. The absorption intensity of macromolecules, such as nucleic acids and proteins, in mildew cells at this wavelength was consistent with the measurements, indicating damage to the cell membrane, leading to the leakage of intracellular substances and the loss of their original active structure [54]. The absorbance curves of *PC*, *TV*, and *AN* in the control group showed no significant changes during the treatment time of 0–120 min. This observation indicated that the cell structure of the microfungi remained intact, and there was no leakage of intracellular substances outside the cell. In contrast, the absorbance values of *PC* and *TV* sharply increased after 30 min of treatment and reached the maximum value, while the absorbance values of *AN* increased significantly after 30 min of treatment and essentially reached the maximum value after 60 min of treatment, in the test groups of MIC and MFC. The absorbance curves of bamboo mildews treated with either MIC or MFC remained mostly flat after 30 or 60 min, indicating that the ionic liquid of dodecylpyridinium chloride could cause damage to mildew cells in a short period and cause their contents to leak outward. Furthermore, the degree of damage to the mildews was greater with increasing concentration of the solution. In Figure 7, the extracellular fluid of *AN* did not reach its maximum absorbance value within 30 min, which was consistent with the observation of the mycelium of MIC- and MFC-treated microfungi. This phenomenon was attributed to the dense mycelial structure of *AN*, which hindered the entry of the dodecylpyridinium chloride solution in a short period.

### 2.6. Effect of Ionic Liquids on the Extracellular pH of Bamboo Mycorrhizal Fungi

The extracellular solutions of *PC*, *TV*, and *AN*, after being subjected to different concentrations of dodecylpyridinium chloride for 0 min, 30 min, 60 min, 90 min, and 120 min, were analyzed for pH values using a micro-pH meter. The findings are presented in Figure 8.

Intracellular processes such as DNA and RNA transcription, protein synthesis, and enzyme activity require a stable pH environment, and pH plays a crucial role in maintaining the intracellular pH balance [55,56]. Under normal conditions, the fluid environment outside the cell is typically neutral, while the fluid environment inside the cell is acidic. However, when the cell structure is affected or damaged, protons start to flow outward, resulting in an increase in external pH, while the intracellular proton discharge is reduced to maintain the acid–base balance within the cell [57].

In Figure 8, the pH values of the extracellular fluid of bamboo mycorrhizal fungi in the control and experimental groups exhibited a trend of initial decrease, followed by an increase, and then a further decrease during the treatment period of 0–120 min. After 30 min of treatment, the pH of *PC*, *TV*, and *AN* in both control and experimental groups decreased to below 6.8, with the pH decrease being more pronounced in the control group compared to the MIC and MFC groups. The pH values of *PC*, *TV*, and *AN* in both control and experimental groups decreased again during the subsequent 60 min of treatment.

During the time period of 0–30 min, the extracellular fluid pH decreased as a result of external environmental effects and acid secretion by the microfungal cells. However, the membrane structure of the microfungal cells treated with dodecylpyridinium chloride MIC and MFC was damaged to some extent, resulting in a small amount of intracellular proton efflux and a slow decrease in extracellular fluid pH. During the time period of 30–60 min, acid production by the microfungal cells decreased as the extracellular fluid pH dropped to a lower value, thereby regulating the acid–base balance of the inner and outer cells, which is consistent with the findings of Zhang et al. [40]. After 60 min of treatment, both the control and experimental groups exhibited a significant decrease in pH, which can be attributed to the prolonged exposure of mildew cells to the external environment and the subsequent increase in acid production [58]. It can be inferred that the dodecylpyridinium chloride ionic liquid can disturb or disrupt the pH balance of the extracellular fluid to some extent, which can inhibit or kill microfungi.

## 3. Materials and Methods

### 3.1. Materials

The following chemical compounds and microbial strains were utilized in the study: 1-decyl-3-methylimidazolium chloride (DMIC, 99% purity) and benzyldimethyldodecylammonium chloride (DDBAC, 99% purity) were used as analytical reagents, while dodecylpyridinium chloride (DPC, 98% purity) and pentanediol (50% purity) were procured from Shanghai Maclean Biochemical Technology Co. Ltd. (Shanghai, China) The microbial strains employed in the study were *Penicillium citrinum* (*PC*), *Trichoderma viride* (*TV*), *Aspergillus niger* (*AN*), and Mix (a mixture comprising equal proportions of *PC*, *TV*, and *AN* spore suspensions).

### 3.2. Preparation of Potato Dextrose Agar (PDA)

In this study, the preparation of potato glucose agar medium involved washing and peeling potatoes weighing 200 g, adding an appropriate quantity of deionized water, and boiling them for 30 min. The juice was subsequently filtered through two layers of gauze. Then, 24 g of agar powder and 20 g of anhydrous glucose were weighed and sequentially added to the filtrate in a beaker. Deionized water was added to the mixture, and it was stirred thoroughly until the agar powder and glucose had completely dissolved. The resulting solution was adjusted to a final volume of 100 mL with deionized water. The prepared medium solution was further divided into wide-mouth conical flasks and subjected to autoclaving (MLS-3750) at 121 °C and 0.1 MPa for 30 min. Finally, the potato glucose agar medium was used for mildew inoculation and culture.

### 3.3. Preparation of Microfungal Suspensions

To prepare a microfungal suspension, a wide-mouth bottle was filled with an appropriate quantity of deionized water and 15–20 small glass beads. The bottle mouth was sealed with a high-temperature-resistant sterile film and, together with the inoculation needle, subjected to autoclaving at 121 °C and 0.1 MPa for 30 min. The appropriate quantity of mycelium or spores was picked or scraped from the surface of the strain and placed into the sterile water contained in the wide-mouth bottle. Subsequently, the bottle was tightly capped and vigorously shaken for 5–10 min. This step was carried out under sterile conditions in a BHC-130011 biosafety cabinet sterilized by a UV lamp. The resulting microfungal suspension was then used for the subsequent microfungal strain culture.

### 3.4. Formulation of Ionic Liquids of Different Concentrations

In this experiment, a 1-decyl-3-methylimidazolium chloride solution with a concentration of 40 mg/mL was prepared by dissolving 4 g of the compound in a beaker with an appropriate quantity of deionized water. The resulting solution was poured into a 100 mL volumetric flask and adjusted to a final volume of 100 mL with deionized water. Using the multiplicative dilution method, a series of concentration gradients for 1-decyl-3-methylimidazolium chloride, namely 2.5 mg/mL, 5 mg/mL, 10 mg/mL, 20 mg/mL, and 40 mg/mL, were prepared in six separate test tubes. Similarly, concentration gradients for benzyldimethyldodecylammonium chloride and dodecylpyridinium chloride were prepared using the same method.

### 3.5. Oxford Cup Method to Investigate the Inhibition Performance of Ionic Liquids against Bamboo Mildew

To investigate the inhibitory performance of 1-decyl-3-methylimidazole chloride, benzyldimethyldodecylammonium chloride, and dodecylpyridinium chloride against bamboo mycorrhizal fungi, the Oxford cup method was employed [59]. First, 100 µL of fungal suspension was evenly spread on the pre-prepared plate medium using an applicator. Next, an Oxford cup with dimensions of 8 mm outer diameter and 6 mm inner diameter was placed at the center of a 90 mm diameter Petri dish. Then, 100 µL of varying concentrations of ionic liquids were introduced into the Oxford cup. Finally, the Petri dishes were sealed with a sterile sealing film and incubated in a POX-600A-12H artificial climate incubator at 28 °C and 85 ± 5% humidity for 2–3 days. The diameter of the inhibition zone of each ionic liquid was measured using the crossover method. Each concentration was repeated three times, and the results were averaged to determine the inhibition rate of each mildew. The control group was treated with deionized water. Formula (1) was used to calculate the inhibition rate of the ionic liquid on each mildew.
(1)$$R=\frac{\left(D_{1}-D_{0}\right)}{D_{0}}\times 100\%$$

In the formula, *R* is the inhibition rate, %; *D*_0_ is the diameter of inhibition zone in the control group, mm; *D*_1_ is the diameter of the inhibition zone in the treatment group, mm.

### 3.6. Minimal Inhibitory Concentration (MIC) and Minimal Fungal Concentration (MFC) Determination

The PDA medium and sterilization procedures were consistent with those outlined in Section 3.2 and Section 3.3. The MIC and MFC values of dodecylpyridinium chloride were determined using the multiplicative dilution method [60]. Initially, concentrations of dodecylpyridinium ionic liquid at 2.5 mg/mL, 5 mg/mL, 10 mg/mL, 20 mg/mL, and 40 mg/mL were prepared. Next, 2.5 mg/mL of the ionic liquid was mixed with 100 mL of liquid medium and stirred thoroughly, and three medium plates containing dodecylpyridinium chloride were prepared with equal volumes. These plates were allowed to cool, and the remaining concentrations of dodecylpyridinium chloride media plates were prepared as described in Section 2.1. A pipette gun was used to evenly coat the plate media with 100 µL of microfungal suspension, and the Petri dishes were sealed with sterile sealing film and incubated for 2 days at 28 °C and 85 ± 5% humidity in an artificial climate incubator. The concentration of the fungi that did not grow on the surface of the medium indicated the minimum inhibitory concentration (MIC) of dodecylpyridinium chloride for the corresponding fungi. Based on the culture of MIC, the fungi at each concentration were incubated for another 7 days, and the concentration of fungi that did not grow on the surface of the medium was used to determine the minimum fungicidal concentration (MFC) of dodecylpyridinium chloride for the corresponding fungi. A control group using deionized water was included in the study.

### 3.7. Effect of Ionic Liquid on Mycelial Morphology of Bamboo Mycorrhizal Fungi

The impact of dodecylpyridinium chloride ionic liquid on the mycelial morphology of bamboo mycorrhizal fungi was investigated using a cold field emission scanning electron emission microscope (SEM) SU8010 [54]. Dodecylpyridinium chloride solutions were prepared at the MIC and the minimum fungal concentration. Multiple 8 mm diameter cakes were obtained from *PC*, *TV*, and *AN* that were cultured for 7 days and had good growth; inoculated on medium plates containing dodecylpyridinium chloride at concentrations of 0, MIC, and MFC; sealed with sterile sealing film; placed in an artificial climate incubator at a temperature of 28 °C and a humidity of 85 ± 5%. After 4 days, the cakes were removed as experimental samples for electron microscopy. Then, the samples were first fixed with 2.5% glutaric acid solution for 12 h at 4 °C and then rinsed with Phosphate-buffered solution (PBS) at pH = 7.0 and 0.1 mol/L for 15 min each time for 3 times. After rinsing, the samples were fixed with 1% osmium acid solution for 2 h, and then rinsed with PBS solution 3 times after fixation. The samples were then dehydrated in different concentrations of ethanol (30%, 50%, 70%, 80%, 90%, 95%) in a gradient of 15 min each, and finally in anhydrous ethanol for 20 min. The freeze-drying machine (DF6006HT) was used for 48 h. Finally, the samples were sprayed with gold and the mycelial morphology of each mildew was observed under SEM.

### 3.8. Effect of Ionic Liquids on the Microstructure of Bamboo Mildew

The microstructure of microfungal cells was investigated using a Transmission Electron Microscope (TEM) JEM-1200. The MIC and MFC concentrations of dodecylpyridinium chloride and the pre-treatment of TEM samples followed the same protocol as described in Section 3.7. However, after the gradient ethanol dehydration treatment, the samples were treated with pure acetone for 20 min. The samples were then permeabilized, embedded, sectioned, and stained before being observed under TEM to analyze the microstructure of each microfungal cell.

### 3.9. Effect of Ionic Liquids on the Release of Cellular Components from Bamboo Mildew

In this study, we investigated the impact of dodecylpyridine ionic liquid on the release of intracellular components of microfungi at MIC and MFC concentration levels using the method described by Paul [61]. *PC*, *TV*, and *AN* microfungal spores that had been cultured for 7 days and had demonstrated satisfactory growth were rinsed with PBS solution at pH = 7.0 and a concentration of 0.1 mol/L, and then suspended in buffer solution. The spores were treated with dodecylpyridinium chloride solution at concentrations of 0, MIC, and MFC for varying durations of 0 min, 30 min, 90 min, and 120 min. After treatment, 5 mL of the spore suspension was taken and centrifuged at 12,000 r/min for 5 min using a Neofuge 18R high-speed centrifuge. The supernatant was collected and its absorbance value at a wavelength of 260 nm was measured on an ultraviolet spectrophotometer (UV-1800), with the control group corrected with PBS solution. Each group was measured three times to ensure the reliability of the results.

### 3.10. Effect of Ionic Liquids on the pH of Extracellular Fluid of Bamboo Mycorrhizal Fungi

The alteration in pH of the extracellular fluid of *PC*, *TV*, and *AN* fungal spores was evaluated following exposure to dodecylpyridinium chloride at concentrations of 0, MIC, and MFC using a PHS-3C micro-pH meter. The method for treatment of the fungal spore suspension was carried out as described in Section 3.9. Specifically, 5 mL of the treated fungal spore suspension was analyzed to determine the pH of the extracellular fluid, with the control group being treated with deionized water. The pH measurements were performed in triplicate for each group.

## 4. Conclusions

This study compared the antimicrobial performance, MIC, MFC, and inhibition mechanisms of three ionic liquids, 1-decyl-3-methylimidazolium chloride, benzyldimethyldodecylammonium chloride, and dodecylpyridinium chloride, against *PC*, *TV*, *AN*, and Mix. Among them, dodecylpyridinium chloride showed the most significant inhibition against all four microorganisms, and its MIC and MFC values were found to be lowest. When *PC* and *TV* were treated with dodecylpyridinium chloride at MIC and MFC concentrations, their extracellular fluid absorbance reached the maximum value after 30 min. The pH of the extracellular fluid of *PC*, *TV*, and *AN* showed a decreasing trend before increasing and then decreasing again between 0 and 120 min, and the ionic liquid caused some damage to the fungal cell membrane structure, leading to a gradual release of protons and a slower decrease in extracellular fluid pH. In summary, dodecylpyridinium chloride exhibited the best antimicrobial activity and damage to fungal cell structures, and the higher the concentration of the ionic liquid, the greater the damage.

## Figures and Tables

**Figure 1 molecules-28-03432-f001:**
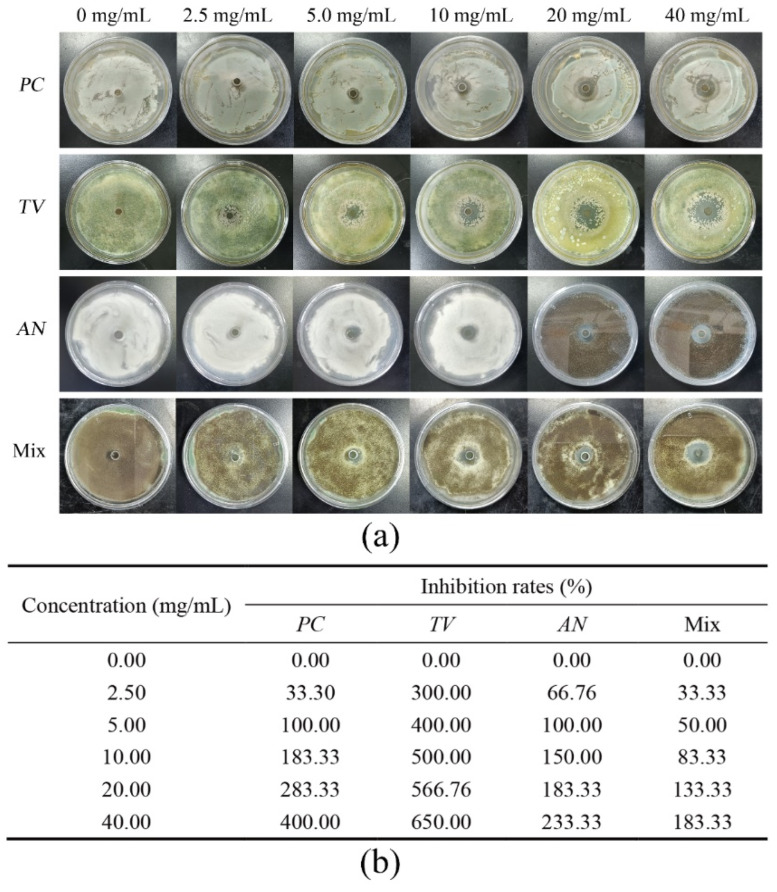
Results of inhibition zone and inhibition rate of different 1-decyl-3-methylimidazole chloride concentrations against bamboo mildew. (**a**) Inhibition zone results; (**b**) inhibition rates.

**Figure 2 molecules-28-03432-f002:**
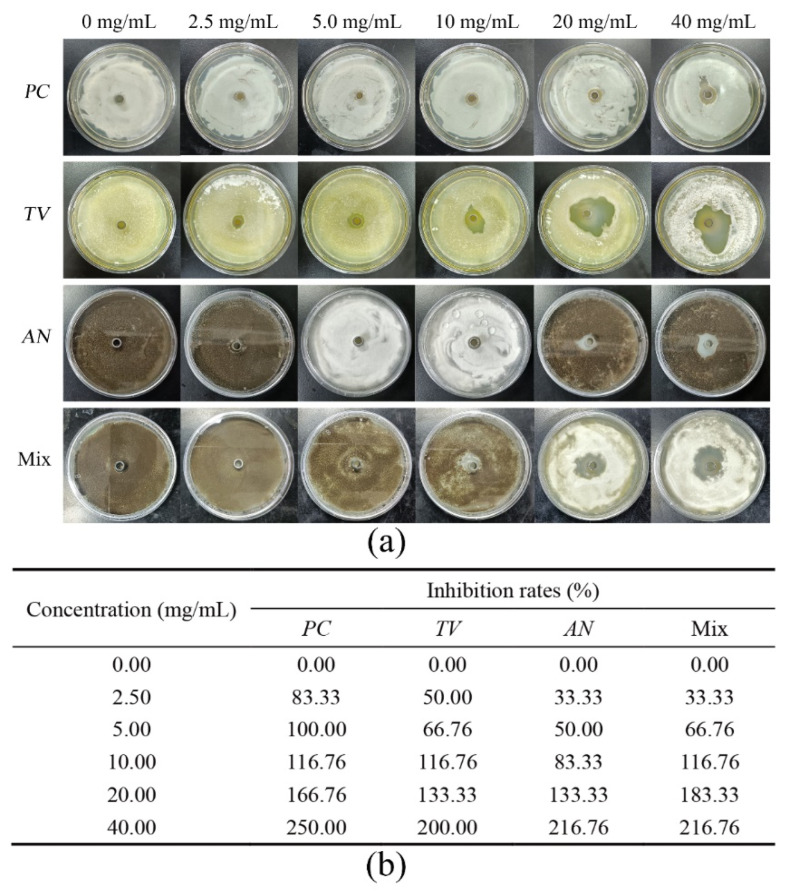
Results of inhibition zone and inhibition rate of different benzyldimethyldodecylammonium chloride concentrations against bamboo mildew. (**a**) Inhibition zone results; (**b**) inhibition rates.

**Figure 3 molecules-28-03432-f003:**
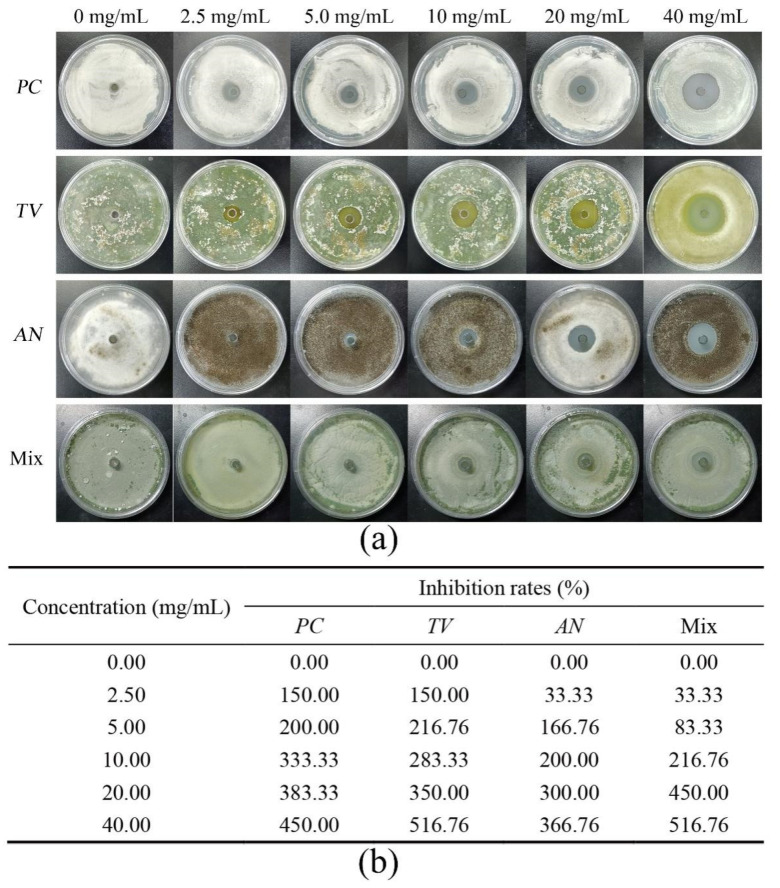
Results of inhibition zone and inhibition rate of different dodecylpyridinium chloride concentrations against bamboo mildew. (**a**) Inhibition zone results; (**b**) inhibition rates.

**Figure 4 molecules-28-03432-f004:**
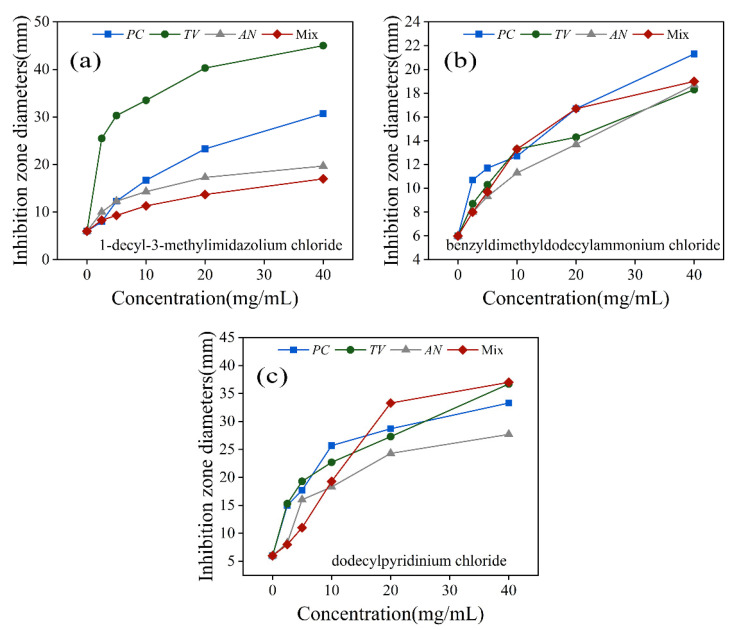
Inhibition zone diameters of three ionic liquids against *PC*, *TV*, *AN*, and Mix. (**a**) 1-decyl-3-methylimidazole chloride; (**b**) benzyldimethyldodecylammonium chloride; (**c**) dodecylpyridinium chloride.

**Figure 5 molecules-28-03432-f005:**
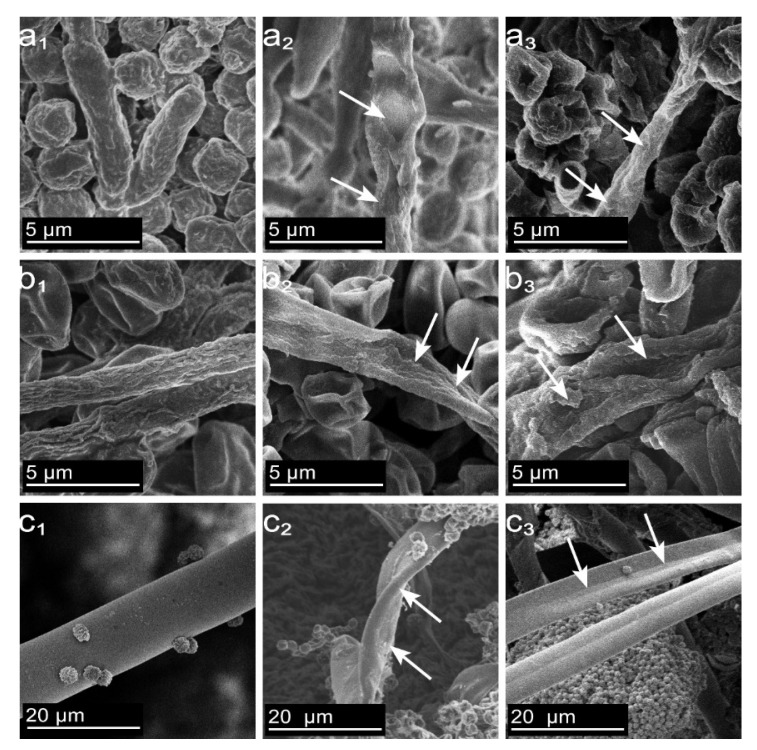
Effect of dodecylpyridinium chloride treatment on the surface morphology of mycelium of bamboo material ((**a**–**c**) PC, TV, and AN, respectively; subscripts (**1**–**3**) are control, MIC, and MFC, respectively).

**Figure 6 molecules-28-03432-f006:**
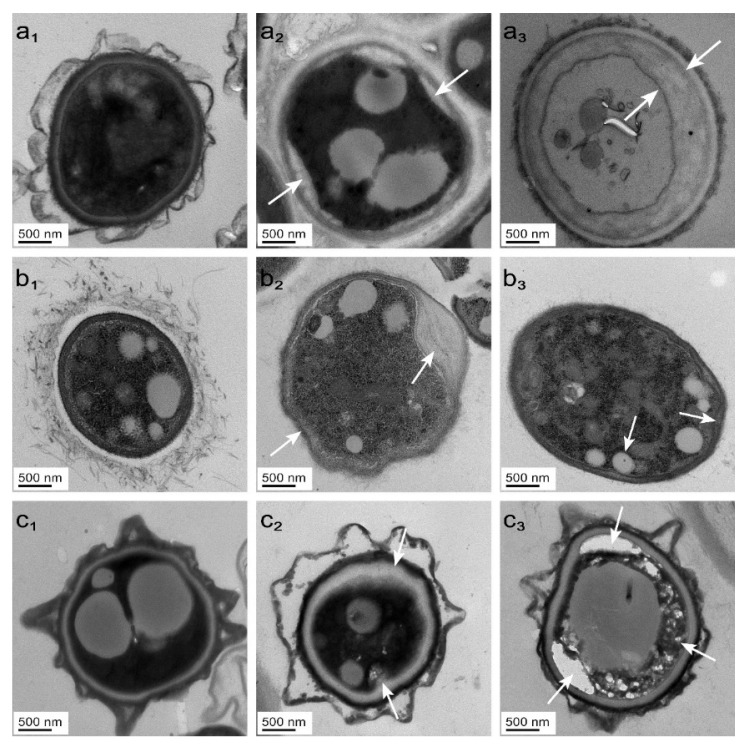
Effect of dodecylpyridinium chloride treatment on the microscopy of bamboo mycorrhizal cells ((**a**–**c**) PC, TV, and AN, respectively; subscripts (**1**–**3**) are control, MIC, and MFC, respectively).

**Figure 7 molecules-28-03432-f007:**
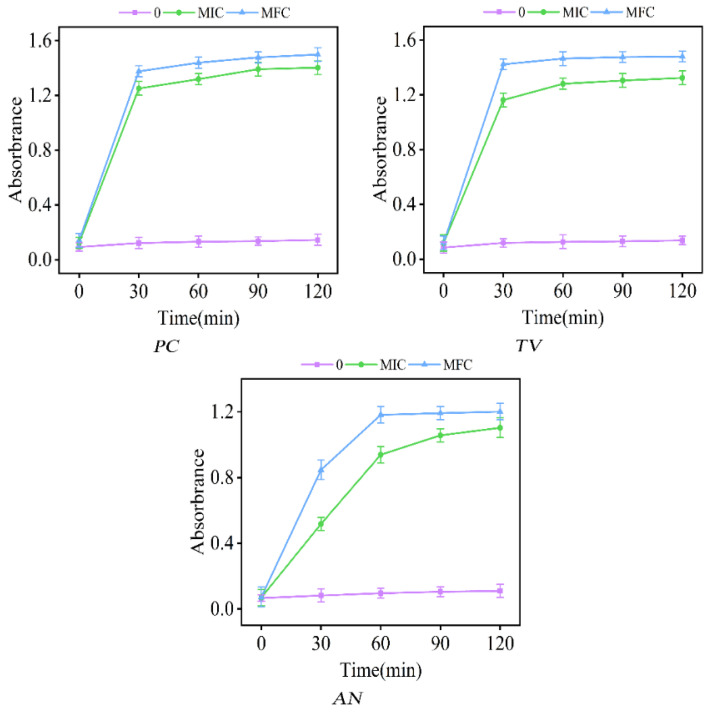
Effect of dodecylpyridinium chloride on the absorbance of 260 nm of the extracellular fluid of bamboo mycorrhizal fungi (control: 0; treated group: MIC, MFC).

**Figure 8 molecules-28-03432-f008:**
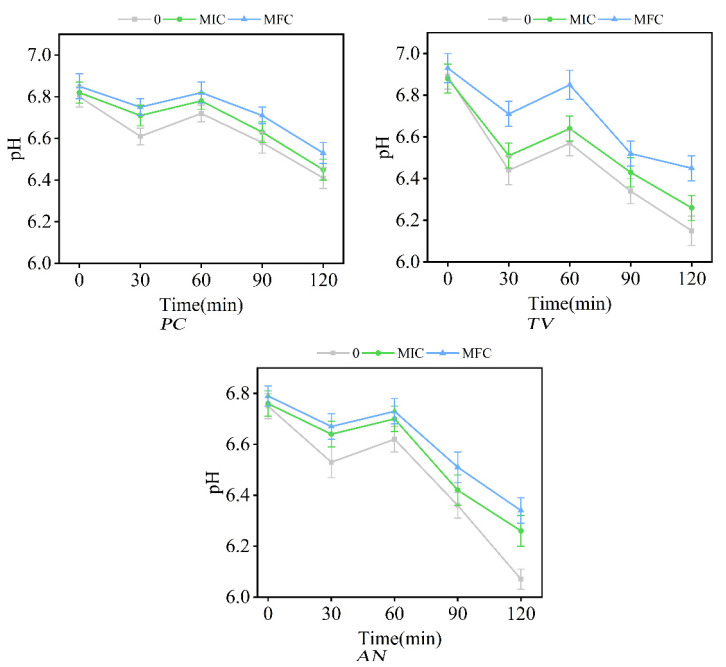
Effect of dodecylpyridinium chloride on pH of extracellular fluid of bamboo mycorrhizal fungi (control: 0; treated group: MIC, MFC).

**Table 1 molecules-28-03432-t001:** MIC and MFC values of different types and concentrations of ionic liquids against bamboo mildew.

Concentration (mg/mL)	MIC (mg/mL)				MFC (mg/mL)			
	*PC*	*TV*	*AN*	Mix	*PC*	*TV*	*AN*	Mix
1-Decyl-3-methylimidazole chloride	10.15	2.61	10.08	20.17	20.09	10.21	20.13	40.15
Benzyldimethyldodecylammonium chloride	5.31	10.23	5.36	5.20	20.24	20.12	40.25	20.10
Dodecylpyridinium chloride	5.37	5.05	5.10	5.23	20.06	10.05	20.10	20.22

## Data Availability

Not applicable.

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
