# Peer review of "Screening of Ionic Liquids against Bamboo Mildew and Its Inhibition Mechanism"

_molecules, 2023, doi:10.3390/molecules28083432_

Round 1

Reviewer 1 Report

The work deals with the study of the antimicrobial properties of three commercially available ionic liquids against Penicillium citrinum, Trichoderma viride, Aspergillus noger. The use of ionic liquids as potential antimicrobial agents is still an open research area. Therefore, the work could be of interest for the readers of the Journal.

However, some parts need to be improved.

In particular, references provided (1,2) about the peculiar properties of ionic liquids should be replaced with more pertinent ones. For instance https://doi.org/10.1021/ie5009597 for the thermal stability, https://doi.org/10.1016/j.molliq.2021.115892 for the low vapor pressure, https://doi.org/10.1021/acs.jpcb.6b04309 for solvent ability for instance towards cellulose.

Furthermore, some claims should be changed. Indeed, often the widespread use of ionic liquids is limited by their high viscosity (1-Decyl-3-methylimidazolium chloride is reported as viscous liquid or solid by Sigma Aldrich) and by their toxicity. The authors should adjust the introduction accordingly. In particular, the authors should explain what is meant with "Ionic liquids possess certain viscosity properties" page 10, and should pay extra attention on the toxicity of the selected structures, considering the potential end use of the ionic liquids in this manuscript.

Author Response

We thank the reviewers for their acknowledgment of the work done in our study, and we respond to the issues in the study accordingly below.

Point 1: In particular, references provided (1,2) about the peculiar properties of ionic liquids should be replaced with more pertinent ones. For instance https://doi.org/10.1021/ie5009597 for the thermal stability, https://doi.org/10.1016/j.molliq.2021.115892 for the low vapor pressure, https://doi.org/10.1021/acs.jpcb.6b04309 for solvent ability for instance towards cellulose.

Response 1: Thank you for your valuable comments and suggestions on the article, pointing out the weak relevance of the literature cited in the first sentence of the introduction describing the properties of ionic liquids. As suggested by the reviewer, the literature describing the properties of ionic liquids in the introduction has been replaced, and the literature has been marked in the place of the corresponding property description. The content has been revised as follows:

In recent years, ionic liquids have garnered significant attention as a "green solvent" due to their unique physicochemical properties, such as low vapor pressure[1], high thermal stability[2], high solubility[3], high viscosity[4], and tunable structures.

  1. Barulli, L.; Mezzetta, A.; Brunetti, B.; Guazzelli, L.; Ciprioti, S.V.; Ciccioli, A. Evaporation thermodynamics of the tetraoctylphosphonium bis (trifluoromethansulfonyl) imide ([P8888] NTf2) and tetraoctylphosphonium nonafluorobutane-1-sulfonate ([P8888] NFBS) ionic liquids. Journal of Molecular Liquids 2021, 333, 115892.
  2. Cao, Y.; Mu, T. Comprehensive investigation on the thermal stability of 66 ionic liquids by thermogravimetric analysis. Industrial & engineering chemistry research 2014, 53, 8651-8664.
  3. Minnick, D.L.; Flores, R.A.; DeStefano, M.R.; Scurto, A.M. Cellulose solubility in ionic liquid mixtures: temperature, cosolvent, and antisolvent effects. The Journal of Physical Chemistry B 2016, 120, 7906-7919.

Point 2: Furthermore, some claims should be changed. Indeed, often the widespread use of ionic liquids is limited by their high viscosity (1-Decyl-3-methylimidazolium chloride is reported as viscous liquid or solid by Sigma Aldrich) and by their toxicity. The authors should adjust the introduction accordingly. In particular, the authors should explain what is meant with "Ionic liquids possess certain viscosity properties" page 10, and should pay extra attention on the toxicity of the selected structures, considering the potential end use of the ionic liquids in this manuscript.

Response 2: Thank you for your valuable comments and suggestions on the article, pointing out the problem of incorrect description of ionic liquid viscosity on page 10 of the text, as well as the problem of potential end-use and toxicity of ionic liquids. Based on the reviewers' suggestions: the description of ionic liquid viscosity in the introduction and page 10 of the text has been revised and the corresponding references have been added, in addition, the potential end-use and toxicity of ionic liquids have been revised as follows:

In recent years, ionic liquids have garnered significant attention as a "green solvent" due to their unique physicochemical properties, such as low vapor pressure[1], high thermal stability[2], high solubility[3], high viscosity[4], and tunable structures.

Ionic liquids possess certain high viscosity[4] properties that enable them to adhere to the surface of cell membranes, facilitating the insertion of hydrophobic alkyl chains into the cell membrane.

  1. Sanchora, P.; Pandey, D.K.; Kagdada, H.L.; Materny, A.; Singh, D.K. Impact of alkyl chain length and water on the structure and properties of 1-alkyl-3-methylimidazolium chloride ionic liquids. Physical Chemistry Chemical Physics 2020, 22, 17687-17704.

The presence of a certain degree of toxicity within the ionic liquid constrains its potential applications. However, the ionic liquid antifungal agent exhibits promising potential in medical, antifungal materials, and other related fields.

Reviewer 2 Report

The manuscript entitled “Screening of Ionic Liquids Against Bamboo Mildew and Its Inhibition Mechanism” presents comprehensive research on antifungal properties of three chosen ionic liquids.

Overall, the manuscript is cohesive, the experiments are well planned, and their results are extensively described. However, the are a few things, that in my opinion, should be adjusted.

The discrepancy in the ionic liquid name, that is benzyldimethyldodecylammonium chloride. Throughout the paper I have found different spellings of the name:

benzyldimethyldodecy lammonium chloride

Dodecyl-dimethyl-benzyl-ammonium chloride

benzyldimethyldodecylammonium chloride

benzyl-dimethyl-dodecyl-ammonium chloride

The authors should pay more detailed attention to the chemical names of ionic liquids, they cannot be changed based on a whim. The nomenclature dictates the proper way to write chemical names of ionic liquids.

Furthermore, Figure 4 is way to large and it is very difficult to read properly. I would suggest reworking this graph, so it would be more esthetically pleasing and clear for the readers.

Detailed revision of the manuscript is required in order to find and correct any mistakes that are present in the current version. Below you can find some examples:

Zhang et al [17]. evaluated six new β-pinene quaternary ammonium salts for their antifungal, antifungal, and anticancer activities against fungi and bacteria, and showed that the novel β-pinene quaternary ammonium salts may cause abnormalities in fungal mycelium, alter cell membrane permeability, and inhibit intracellular ATP activity.

The strength of inhibition was characterized by measuring the diameter of the inhibition zone, and three ionic liquids, namely 1-decyl-3-methylimidazole chloride, benzyldimethyldodecy lammonium chloride, and dodecylpyridinium chloride, were evaluated.

As depicted in Figure 5, PC, TV, and AN all displayed intact mycelial structures with full and uniform thickness in Figure 5(a1), 4(b1), and 4(c1), respectively.

Author Response

We thank the reviewers for their acknowledgement of the work done in our study, and we respond to the issues in the study accordingly below.

Point 1: The discrepancy in the ionic liquid name, that is benzyldimethyldodecylammonium chloride. Throughout the paper I have found different spellings of the name:

benzyldimethyldodecy lammonium chloride

Dodecyl-dimethyl-benzyl-ammonium chloride

benzyldimethyldodecylammonium chloride

benzyl-dimethyl-dodecyl-ammonium chloride

Response 1: Thank you for your valuable comments and suggestions on the article, pointing out the inconsistency in the name of benzyldimethyldodecylammonium chloride in the article, according to your suggestion: the name of benzyldimethyldodecylammonium chloride was unified in several places in the article, and its name was based on the English name given by Maclean's, Benzyldimethyldodecylammonium chloride. The amendments are as follows:

Materials: benzyldimethyldodecylammonium chloride (DDBAC, 99% purity) were used as ana-lytical reagents.

Results and Analysis 3.1: The strength of inhibition was characterized by measuring the diameter of the inhibition zone, and three ionic liquids, namely 1-decyl-3-methylimidazole chloride, benzyldimethyldodecylammonium chloride, and dodecylpyridinium chloride, were evaluated.

Concentration (mg/mL)

MIC (mg/mL)

MFC (mg/mL)

PC

TV

AN

Mix

PC

TV

AN

Mix

1-Decyl-3-methylimidazole chloride

10.15

2.61

10.08

20.17

20.09

10.21

20.13

40.15

Benzyldimethyldodecylammonium chloride

5.31

10.23

5.36

5.20

20.24

20.12

40.25

20.10

Dodecylpyridinium chloride

5.37

5.05

5.10

5.23

20.06

10.05

20.10

20.22

Results and Analysis 3.2: Benzyldimethyldodecylammonium chloride, on the other hand, exhibited a lower MFC value than AN in the Mix environment, likely due to AN not gaining a growth advantage in competition.

Conclusion: This study compared the antimicrobial performance, MIC, MFC, and inhibition mechanisms of three ionic liquids, 1-decyl-3-methylimidazolium chloride, benzyldimethyldodecylammonium chloride, and dodecylpyridinium chloride, against PC, TV, AN, and Mix.

Benzyldimethyldodecylammonium chloride: http://www.macklin.cn/search/Benzyldimethyldodecylammonium%20chloride

Point 2: The authors should pay more detailed attention to the chemical names of ionic liquids, they cannot be changed based on a whim. The nomenclature dictates the proper way to write chemical names of ionic liquids.

Response 2: Thank you for your valuable comments and suggestions on the article. The names of the three ionic liquids used in the article have been standardized based on the reviewers' suggestions, and the English names of the ionic liquids are based on the names provided on the Aladdin and Macklin's reagent websites. Below are the URLs for the English names of 1-decyl-3-methylimidazole chloride, benzyldimethyldodecylammonium chloride, and dodecylpyridinium chloride.

1-Decyl-3-methylimidazole chloride: http://www.macklin.cn/products/D856806

Benzyldimethyldodecylammonium chloride: http://www.macklin.cn/search/Benzyldimethyldodecylammonium%20chloride

Dodecylpyridinium chloride: http://www.macklin.cn/products/D823218

Point 3: Furthermore, Figure 4 is way to large and it is very difficult to read properly. I would suggest reworking this graph, so it would be more esthetically pleasing and clear for the readers.

Response 3: Thank you for your valuable comments and suggestions on the article. Figure 4 shows a graph comparing the diameter of the inhibition circles of three ionic liquids against mycobacteria. At the suggestion of the reviewer, we have made three copies of this graph for comparison. The title text and scale size of the horizontal and vertical coordinates of the graph were also adjusted accordingly and the redrawn results are as follows:

Figure 4 Inhibition zone diameters of three ionic liquids against PC, TV, AN and Mix ((a): 1-decyl-3-methylimidazole chloride; (b): benzyldimethyldodecylammonium chloride; (c): dodecylpyridinium chloride).

It can be seen from Figure 4 that at the same concentration, 1-decyl-3-methylimidazolium chloride in Figure 4(a) showed a larger circle of inhibition against TV at the same concentration; benzyl dimethyl ammonium chloride in Figure 4(b) showed a smaller circle of inhibition against all strains tested; and dodecylpyridinium chloride in Figure 4(c) showed a larger circle of inhibition against all strains tested, with an inhibition diameter of 17 mm against AN at 10 mg/mL.

Point 4: Detailed revision of the manuscript is required in order to find and correct any mistakes that are present in the current version. Below you can find some examples:

Zhang et al [17]. evaluated six new β-pinene quaternary ammonium salts for their antifungal, antifungal, and anticancer activities against fungi and bacteria, and showed that the novel β-pinene quaternary ammonium salts may cause abnormalities in fungal mycelium, alter cell membrane permeability, and inhibit intracellular ATP activity.

Response 4: Thank you for your valuable comments and suggestions on the article, which pointed out problems with the citations of antimicrobial studies in the article and provided us with directions for revising the article. Based on the reviewers' suggestions, we have revised the citations in the introduction for antimicrobial studies done by other researchers to make the citations more accurate. In addition, the descriptions of Figures 7 and 8 in the article were revised and the coordinate scale of Figure 8 was adjusted to make it more readable, with the following changes.

Dani et al. [36] The interfacial behavior of surface active ionic liquids (SAILs) was verified in combination with biocidal and cytotoxic assessments. The results revealed that SAILs with the alkyl chain-length greater than C-8- act as a fair antimicrobial agent against the selected microbial strain which is attributed to the enhanced degree of SAILs hydrophobicity. Zhang et al [19]. The antibacterial, antifungal and anticancer properties of β-pinene quaternary ammonium salts were designed, synthesised and evaluated. The results showed that bis-hydronopyl dimethyl ammonium chloride and bis-hydronopyl dimethyl ammonium iodide presented remarkable antimicrobial activity against the tested fungi and bacteria. Rezki et al. [37] A number of pyridine-based ionic liquids: novel inhibitors of fungal ergosterol biosynthesis were investigated and the results showed that some of these pyridine-based ionic liquids have significant anti-Candida activity, possibly by interacting with ergosterol in the fungal cell membrane to reduce its content and ultimately lead to membrane damage. Trush et al. [38] The ecotoxicity and degradability of cationic fungicides with ester-functionalised pyridinium ionic liquids was studied. The results showed that both ester-functionalised ILs and cetylpyridinium chloride (CPC) showed significantly reduced antibacterial activity compared to 1-dodecylpyridinium chloride (PyrC(12)-Cl), but the ester-functionalised ILs exhibited excellent antifungal activity against Candida albicans.

  1. Dani, U.; Bahadur, A.; Kuperkar, K.; Safety, E. Validating interfacial behaviour of surface-active ionic liquids (SAILs) with computational study integrated with biocidal and cytotoxic assessment. Ecotoxicology 2019, 186, 109784.
  2. Zhang, L.; Feng, X.-Z.; Xiao, Z.-Q.; Fan, G.-R.; Chen, S.-X.; Liao, S.-L.; Luo, H.; Wang, Z.-D. Design, synthesis, antibacterial, antifungal and anticancer evaluations of novel β-pinene quaternary ammonium salts. International Journal of Molecular Sciences 2021, 22, 11299.
  3. Rezki, N.; Al-Sodies, S.A.; Shreaz, S.; Shiekh, R.A.; Messali, M.; Raja, V.; Aouad, M.R. Green ultrasound versus conventional synthesis and characterization of specific task pyridinium ionic liquid hydrazones tethering fluorinated counter anions: Novel inhibitors of fungal Ergosterol biosynthesis. Molecules 2017, 22, 1532.
  4. Trush, M.; Metelytsia, L.; Semenyuta, I.; Kalashnikova, L.; Papeykin, O.; Venger, I.; Tarasyuk, O.; Bodachivska, L.; Blagodatnyi, V.; Rogalsky, S.; et al. Reduced ecotoxicity and improved biodegradability of cationic biocides based on ester-functionalized pyridinium ionic liquids. Environmental Science 2019, 26, 4878-4889.

Figure 7 Effect of dodecylpyridinium chloride on the absorbance of 260 nm of the extracellular fluid of bamboo mycorrhizal fungi (control: 0; treated group: MIC, MFC).

Figure 8 Effect of dodecylpyridinium chloride on pH of extracellular fluid of bamboo mycorrhizal fungi (control: 0; treated group: MIC, MFC).

Point 5: The strength of inhibition was characterized by measuring the diameter of the inhibition zone, and three ionic liquids, namely 1-decyl-3-methylimidazole chloride, benzyldimethyldodecy lammonium chloride, and dodecylpyridinium chloride, were evaluated.

Response 5: Thank you for your valuable comments and suggestions on the article, as suggested by the reviewer, the expression of the inhibition strength for ionic liquids may not be in place to cause you problems. In this article, the Oxford cup method was used for the inhibition circle test. The Oxford cup allows the liquid to spread evenly in all directions, ensuring that the final inhibition circle results are more comparable. To compare the size of the inhibition circle at the same concentration, the inhibition rate of the ionic liquids is calculated by the formula R=((D1-D0))/D0 × 100%, thus comparing the inhibition strength of each ionic liquid on a macro scale.

Point 6: As depicted in Figure 5, PC, TV, and AN all displayed intact mycelial structures with full and uniform thickness in Figure 5(a1), 4(b1), and 4(c1), respectively.

Response 6: Thank you for your valuable comments and suggestions on the article. In Figure 5 we have described the mycelial structures of PC, TV and AN in the control group too uniformly and in general terms. Based on the reviewers' suggestions, we have developed a more detailed phenomenological description of the mycelial structures of PC, TV and AN in the control group, which is revised as follows:

According to the observations depicted in Figure 5, the mycelium of PC displays a well-developed and substantial structure, characterized by a scaly sheath-like surface in Figure 5(a1). In contrast, the mycelium of TV exhibits a straight and irregularly patterned surface in Figure 5(b1). Moreover, the mycelium of AN displays a rounded and thick surface with a smooth appearance in Figure 5(c1).

Round 2

Reviewer 1 Report

The authors improved the manuscript following this reviewer's remarks.

Therefore, I recommend this manuscript for publication.